# A Novel Carbohydrate Fatty-Acid Monosulphate Ester, Squalane-in-Water Adjuvant Is Safe and Enhances Inactivated Influenza Vaccine Immunogenicity in Older Adults

**DOI:** 10.3390/vaccines13090922

**Published:** 2025-08-29

**Authors:** Valentino D’Onofrio, Bart Jacobs, Azhar Alhatemi, Simon De Gussem, Marjolein Verstraete, Sharon Porrez, Anthony Willems, Fien De Boever, Gwenn Waerlop, Geert Leroux-Roels, Els Michels, Francesca Vanni, Alessandro Manenti, Peter Paul Platenburg, Luuk Hilgers, Isabel Leroux-Roels

**Affiliations:** 1Center for Vaccinology (CEVAC), Ghent University and Ghent University Hospital, 9000 Ghent, Belgium; valentino.donofrio@uzgent.be (V.D.); azhar.alhatemi@uzgent.be (A.A.); simon.degussem@uzgent.be (S.D.G.); marjolein.verstraete@ugent.be (M.V.); sharon.porrez@ugent.be (S.P.); anthony.willems@ugent.be (A.W.); fien.deboever@uzgent.be (F.D.B.); gwenn.waerlop@ugent.be (G.W.); geert.lerouxroels@ugent.be (G.L.-R.); 2Harmony Clinical Research BV, 9090 Melle, Belgium; e.michels@harmony-cr.com; 3VisMederi S.r.l., 53035 Monteriggioni, Italy; francesca.vanni@vismederi.com (F.V.); alessandro.manenti@vismederi.com (A.M.); 4LiteVax, 4061 BJ Ophemert, The Netherlands; peterpaul.platenburg@litevax.com (P.P.P.); luuk.hilgers@litevax.com (L.H.)

**Keywords:** adjuvant, influenza vaccine, immunogenicity, safety, reactogenicity, older adults

## Abstract

Influenza vaccines are the primary strategy to prevent severe influenza disease; however, their efficacy is often suboptimal, particularly in older adults (OAs). LiteVax Adjuvant (LVA), a novel adjuvant containing carbohydrate fatty-acid monosulphate ester (CMS) as the active ingredient, has demonstrated a favourable safety profile and enhanced immunogenicity when combined with a low-dose seasonal influenza vaccine in adults aged 18 to 50 years in a first-in-human phase 1 study. The present study investigates the reactogenicity and immunogenicity of CMS-based adjuvanted seasonal influenza vaccine in OAs, with a comparison to responses in younger adults (YAs). In this phase 1b, double-blind, active-controlled clinical trial, 36 YAs (18–50 years) and 48 OAs (≥60 years) were randomized (1:1:1) to receive either 0.5 mg or 1 mg LVA combined with VaxigripTetra, or VaxigripTetra alone. Solicited adverse events (AEs) were recorded using an electronic diary for 7 days following vaccination. Hemagglutination inhibition (HI) titers against four influenza strains were measured at baseline (pre-vaccination) and at 7-, 28-, and 180-days post-vaccination. All 24 YAs and 31 out of 32 OAs receiving CMS-based adjuvanted vaccines reported pain post-vaccination, compared to 8/12 YAs and 4/16 OAs receiving VaxigripTetra. Systemic AEs were more frequently reported among YAs receiving CMS-based adjuvanted vaccines (22/24) compared to those receiving VaxigripTetra (8/12). In OAs, the number of systemic AEs was similar regardless of CMS-based adjuvant administration. Most AEs were mild to moderate and resolved within 3 days. Both CMS-based adjuvanted formulations elicited increased HI titers at Day 7, peaking at Day 28, with a decline thereafter that remained above baseline at Day 180. In YAs, HI titers were comparable between the CMS-based adjuvanted and non-adjuvanted vaccines across all strains and timepoints. In contrast, CMS-based adjuvanted vaccination in OAs induced higher HI titers at Days 28 and 180 for all influenza strains tested. LVA shows an acceptable safety profile in both age cohorts and enhances humoral immune responses in older adults. The 1 mg dose of LVA was more immunogenic, highlighting its potential utility in this target population. Future research will focus on elucidating the mechanisms underlying the immunostimulatory effect of the CMS-based adjuvant.

## 1. Introduction

Seasonal influenza causes mild or severe illnesses, including hospitalization and, in some cases, death. Older adults (OAs), children under 5 years old, individuals with chronic illnesses, and pregnant women are at higher risk of developing severe illness and complications [1]. The increased susceptibility to influenza in adults aged 60 years and older is partly related to immunosenescence [2,3]. Over time, the effectiveness of the immune system declines, characterized by reductions in the number and diversity of memory B and T cells, while other functions, such as inflammatory cytokine production, may be increased [2,3,4]. Overall, there is an age-dependent remodeling of the immune system.

Annual vaccination remains the most effective approach to prevent serious complications and death caused by influenza. The predominant seasonal vaccines in the 2023–2024 season were tetravalent inactivated vaccines targeting hemagglutinin (HA) and neuraminidase (NA) of four influenza virus strains (two A subtypes and two B lineages) [5]. However, in the context of immunosenescence, vaccines tend to be less immunogenic and reactogenic in older adults compared to younger populations [6,7]. Indeed, vaccine effectiveness (VE) of standard-dose vaccines has been estimated at 50 to 60% in adults aged 60 years and older for preventing laboratory-confirmed influenza [8,9]. Numerous studies have shown decreased humoral and cellular immunogenicity in older adults [6,8,10,11], with reported hemagglutination inhibition (HI) responses up to 8-fold lower after vaccination in this age group [12]. Furthermore, they tend to have lower concentration of IgG antibodies [12,13], with similar avidity and affinity, alongside altered memory CD4+ T cell response [14].

Hence, there is a clear need for more effective influenza vaccines. A new experimental vaccine against seasonal influenza virus infection has been developed, combining a full dose of a licensed, tetravalent, seasonal, inactivated split virion influenza virus vaccine (VaxigripTetra) with LiteVax Adjuvant (LVA)—A novel, potent adjuvant [15]. LVA belongs to a new class of synthetic carbohydrate fatty-acid monosulphate esters (CMS), immobilized on the surface of oil droplets within a squalane-in-water nano-emulsion. Several preclinical studies have demonstrated the potent efficacy of single-shot vaccines containing CMS, showing comparable or superior antibody titers compared to two doses of MF59-adjuvanted vaccines [15].

In a phase 1, first-in-human (FIH) study, LVA exhibited an acceptable safety profile in younger adults aged 18 to 50 years. When LVA was added to a low (1/5th) dose of the licensed VaxigripTetra, HI titers and frequencies of polyfunctional CD4+ T cells were comparable to those elicited by a standard-dose non-adjuvanted VaxigripTetra [16].

The goal of the present study was to investigate the reactogenicity and immunogenicity of the CMS-based adjuvant combined with a full-dose seasonal influenza vaccine in OAs. Additionally, we compared immunogenicity outcomes in OAs and younger adults (YAs) to explore LVA’s potential to enhance immune responses and counteract immunosenescence.

## 2. Methods

### 2.1. Study Design and Participants

This study was a randomized, double-blind, active-controlled phase 1 trial conducted at a single center, the Center for Vaccinology (CEVAC, Ghent University Hospital and Ghent University in Ghent, Belgium), between 22 January and 10 October 2024. The study enrolled 84 healthy male and female adults, including 36 aged between 18 and 50 years and 48 over 60 years. Eligible participants had not experienced a confirmed influenza virus infection in the 12 months prior to vaccination and had not received an influenza vaccine within the preceding 6 months. Detailed inclusion and exclusion criteria are available in Appendix A. All study procedures were conducted in accordance with the International Council for Harmonisation (ICH) guidelines for Good Clinical Practice (GCP). The study documents were approved by an independent Ethics Committee and the Belgian Federal Agency for Medicines and Health Products (FAMHP) (EudraCT number: 2023-508230-33-00, NCT06294262). Written informed consent was obtained from all participants prior to any study-related procedures.

### 2.2. Study Vaccine

The adjuvant formulation consisted of CMS combined with a squalane-in-water emulsion. This fully synthetic, sterile, aqueous, ready-for-use product is characterized by its physical and chemical stability. In this study, LVA containing 0.5 mg or 1 mg CMS was mixed with VaxigripTetra (Sanofi Pasteur, Lyon, France), a licensed quadrivalent seasonal inactivated split-virion influenza vaccine, immediately prior to intramuscular administration. Each vaccine dose contained 15 µg of HA per strain and was administered either alone or combined with 0.5 mg or 1 mg of CMS. The 2023–2024 Northern Hemisphere formulation of VaxigripTetra, composed according to WHO’s recommendations for that season, was used in all study groups. The injection volume was 0.5 mL for VaxigripTetra and 0.55 mL for VaxigripTetra combined with LVA.

### 2.3. Study Procedures

Thirty-six (36) healthy YA participants aged 18–50 years were randomized in an observer-blind manner into three cohorts (1:1:1 randomization), with 12 participants each: Cohort YA-CTRL received unadjuvanted VaxigripTetra, Cohort YA-VXT + 0.5 mg CMS received VaxigripTetra + LVA containing 0.5 mg CMS, and Cohort YA-VXT + 1 mg CMS received VaxigripTetra + LVA containing 1 mg CMS. Additionally, 48 healthy OA participants aged 60 years or older were randomized similarly (1:1:1) into three cohorts of 16 participants each: Cohort OA-CTRL received unadjuvanted VaxigripTetra, Cohort OA-VXT + 0.5 mg CMS received VaxigripTetra + LVA containing 0.5 mg CMS, and Cohort OA-VXT + 1 mg CMS received VaxigripTetra + LVA containing 1 mg CMS. The study began with three sentinel YA participants randomized 2:1 into YA-VXT + 0.5 mg CMS and YA-CTRL. After a 24-h safety review, three additional participants were vaccinated using the same 2:1 randomization. The remaining 12 participants (YA-VXT + 0.5 mg CMS: n = 8, YA-CTRL: n = 4) were vaccinated after a 7-day post-vaccination safety review. The same staged approach was applied to YA-VXT + 1 mg CMS and the remaining half of YA-CTRL. Simultaneously, the same procedures were also applied to OA participants in OA-VXT + 0.5 mg CMS and OA-CTRL, and subsequently for OA-VXT + 1 mg CMS. An overview of the study design is shown in Appendix A. All participants were centrally assigned to a study vaccine cohort using an interactive web response system (IWRS) built into the eCRF, which was based on a randomization list prepared by an unblinded statistician. Study vaccines were prepared by an unblinded pharmacist and administered by an unblinded nurse, both of whom were not involved in any other assessments during the whole study. All other procedures, including clinical observations and data collection, were carried out by blinded study personnel. During vaccine administration, participants were asked to turn their head in the opposite direction of the syringe in order not to identify the study vaccine and maintain the double-blind.

Following vaccination, safety monitoring included reporting of solicited and unsolicited adverse events (AEs), serious adverse events (SAEs), adverse events of special interest (AESIs), and potential immune-mediated events (pIMDs). This was combined with physical examinations, vital signs, and standard haematology and biochemistry laboratory tests (Appendix A). Participants used an electronic diary for 7 days post-vaccination to record solicited local and systemic AEs, as well as oral body temperature. Solicited local AEs included redness, swelling, induration, and pain at the injection site. Systemic AEs included fever, fatigue, headache, arthralgia, myalgia, and malaise. All reported AEs were evaluated by the investigator and graded according to the FDA’s toxicity grading scale for preventive vaccine trials in healthy adults and adolescents [17]. Participants were instructed to report any health status changes until 28 days after vaccination (unsolicited AEs). SAEs, AESIs (e.g., anaphylaxis, convulsions), and pIMDs (e.g., autoimmune or inflammatory neurological conditions) were monitored throughout the entire study period. The investigator assessed causality of all reported AEs using clinical judgement. The relationship to the study vaccine was classified as definitely related, probably related, unlikely related, or not related. A reasonable possibility of a relationship included facts, evidence, or arguments to suggest causality. Alternative causes, underlying diseases, concomitant therapy, risk factors, and temporal relationship to study vaccine administration were also considered when defining the causality. All solicited local AEs were considered related to the study vaccine by default.

Serum samples were collected from all participants at baseline (Day 0) and on Days 7, 28, and 180 to determine HI titers and microneutralization (MN) antibody titers against the four influenza vaccine strains. Influenza infection was monitored continuously during the study using nasal self-tests in symptomatic participants to assess the potential impact of intercurrent infections on observed immune responses. Details of the immunological assays and influenza surveillance methods are provided in the Appendix A.

### 2.4. Endpoints

Primary endpoints included the number of participants experiencing at least one solicited AE during the 7-day post-vaccination period (local, systemic, or either) of any severity grade (mild, moderate, severe), the occurrence of unsolicited AEs during the 28-day post-vaccination period, and the presence of SAEs, AESIs, and pIMDs throughout the complete study period within each cohort.

Secondary endpoints were HI and MN antibody titers against the four influenza strains included in VaxigripTetra. These were assessed using serum samples collected at each timepoint. Seroprotection and seroconversion rates were calculated based on HI titers: seroprotection was defined as an HI antibody titer ≥ 40, while seroconversion was defined as a 4-fold increase in HI antibody titer compared to pre-vaccination level.

### 2.5. Statistical Analysis

Categorical variables were summarized, using counts and percentages, while continuous variables were presented as means (arithmetic or geometric), medians, standard deviations (SDs), interquartile ranges (IQRs), 95% confidence intervals (CIs), minimum (min), and maximum (max). Three analysis sets were defined: the entered analysis set (EAS) comprised all participants who signed the informed consent, the safety analysis set (SAS) included all participants who received at least one vaccine dose, and the immunogenicity analysis set (IAS) included all participants in the SAS with at least one post-vaccination blood sample available for immunogenicity assessment. Differences in GMT titers between cohorts at each visit were analyzed using the Wilcoxon rank-sum test, with multiple testing correction via the Benjamini–Hochberg procedure. Differences in seroconversion and seroprotection rates between cohorts at each visit were analyzed using Fisher’s Exact test. For exploratory analysis of antibody responses, linear mixed-effect models were performed on log2-transformed HI and MN titers. A *p*-value < 0.05 was considered statistically significant. All statistical analyses and data visualizations were performed using R (v4.4.1) and RStudio (v2025.05.1+513).

## 3. Results

### 3.1. Study Population Demographics

A total of 148 participants were screened for the study, of whom 84 (56.8%) were randomized (Figure 1). The remaining 64 participants were not assigned to any treatment cohort, including 30 due to screening failure, 29 back-up candidates who were not needed, 3 who withdrew prior to participation, 1 who became ill the night before vaccination, and 1 who took an analgesic within 24 h of the planned dose. All but one of the randomized participants (83/84, 98.8%) completed the study. One participant in OA-VXT + 0.5 mg CMS discontinued after Day 28 but before Day 180 due to an unrelated AE. Additionally, one participant in OA-CTRL was excluded from the immunogenicity analysis because of previously undiagnosed hypothyroidism.

Overall, study participants had a mean age of 49.8 (±19.8) years. YAs across cohorts ranged from 27.8 (±7.7) to 28.9 (±9.0) years, while OAs ranged from 63.6 (±2.6) to 67.2 (±5.0) years (Table 1). The majority were female (48/84, 57.1%) and white (81/84, 96.4%). The mean weight at inclusion was 73.36 (±11.88) kg, with a mean BMI of 24.95 (±3.39) kg/m^2^. No participants were pregnant at the time of enrolment. Additionally, no participants had self-test-confirmed influenza infection during the study.

### 3.2. Reactogenicity

In total, 74/84 (88.1%) participants experienced at least one solicited AE during the 7-day post-vaccination period.

At least one solicited local AE was reported by 67/84 (79.8%) participants, with a higher frequency in YA-VXT + 0.5 mg CMS (12/12, 100.0%), YA-VXT + 1 mg CMS (12/12, 100.0%), OA-VXT + 0.5 mg CMS (16/16, 100.0%), and OA-VXT + 1 mg CMS (15/16, 93.8%), compared to YA-CTRL (8/12, 66.7%) and OA-CTRL (4/16, 25.0%) (Figure 2A). The most common solicited local AE was pain, mostly mild (52/84, 61.9%) or moderate (15/84, 17.9%). All participants in YA-VXT + 0.5 mg CMS, YA-VXT + 1 mg CMS, and OA-VXT + 0.5 mg CMS experienced pain (100.0%), while in OA-VXT + 1 mg CMS, 15/16 participants (93.8%) experienced pain; in YA-CTRL, 8/12 (66.7%) and in OA-CTRL, 4/16 (25.0%). No severe local solicited AEs were reported (Figure 2A).

Overall, 54/84 (64.3%) participants reported solicited systemic AEs, with higher frequencies in YA-VXT + 0.5 mg CMS and YA-VXT + 1 mg CMS (11/12, 91.7% in each) compared to YA-CTRL (8/12, 66.7%), and similar rates across older adult cohorts: OA-CTRL, 9/16 (56.3%), OA-VXT + 0.5 mg CMS, 7/16 (43.8%), and OA-VXT + 1 mg CMS, 8/16 (50.0%) (Figure 2A). Oral body temperature increased only slightly in the evening and on Day 1 post-vaccination. In total, five participants reported mild fever (38.0–38.4 °C) all on Day 1 post-vaccination (Figure 2B). These were three YAs (2/12 (16.7%) in YA-VXT + 0.5 mg CMS and 1/12 (8.3%) in YA-VXT + 1 mg CMS) and two OAs (1/16 (6.3%) in OA-VXT + 0.5 mg CMS and 1/16 (6.3%) in OA-VXT + 1 mg CMS). Oral body temperature decreased to baseline on Day 2 post-vaccination (Figure 2B). Solicited systemic AEs were reported more frequently in YAs than in OAs across all vaccine groups (Figure 2A). The most reported symptoms were fatigue and headache, both typically mild to moderate in severity and resolving within 2–3 days. In YAs, systemic AEs were more frequent in LVA groups (YA-VXT + 0.5 mg CMS and YA-VXT + 1 mg CMS) compared to the unadjuvanted vaccine (YA-CTRL). In contrast, OAs reported fewer systemic AEs overall, and differences between adjuvanted and non-adjuvanted groups were less pronounced.

The mean time to onset of solicited AEs was 0.71 days (95% CI: 0.57–0.85). All solicited AEs resolved spontaneously, with a mean duration of 2.19 days (95% CI: 1.99–2.39).

Reactogenicity exhibited a dose–response relationship to CMS in YAs, with higher average severity during 7 days after vaccination observed with increasing doses (Figure 2C). Although reactogenicity in OAs was higher with LVA compared to controls, it was not dose-dependent (Figure 2C).

### 3.3. Safety

Overall, 65/84 participants (77.4%) reported unsolicited AEs within the 28-day post-vaccination period, with a total of 136 events. A higher proportion of participants in YA-VXT + 0.5 mg CMS and YA-VXT + 1 mg CMS (11/12, 91.7% each) experienced unsolicited AEs compared to YA-CTRL (8/12, 66.7%). In OA-CTRL and OA-VXT + 0.5 mg CMS, 11/16 (68.8%) participants in each cohort reported unsolicited AEs, while in OA-VXT + 1 mg CMS, 13/16 (81.3%) participants did so (Table 2).

Of these, 37/84 participants (44.0%) experienced a total of 65 unsolicited AEs considered related to the study vaccine by the investigator during the 28-day post-vaccination period. Most of these related AEs occurred in YA-VXT + 1 mg CMS (11/12, 91.7%), YA-VXT + 0.5 mg CMS (9/12, 75.0%), and OA-VXT + 1 mg CMS (11/16, 68.8%), compared to OA-CTRL (3/16, 18.8%), YA-CTRL (2/12, 16.7%), and OA-VXT + 0.5 mg CMS (1/16, 6.3%). The most frequently reported related unsolicited AEs were injection site conditions (pain and swelling) (23/84, 27.4%) and pyrexia (11/84, 13.1%), which occurred later than 7 days post-vaccination, which was defined as the period for solicited events (“late-onset reactions”). Both unsolicited AEs were mostly mild and occurred more frequently after vaccination with LVA. Five participants (5/84, 6.0%) reported Grade 3 unsolicited AEs, with a total of seven events, including only “injection site swelling” and “injection site erythema” as related to the study vaccine. No participants experienced unsolicited AEs of Grade 4.

Four participants (4.8%) experienced SAEs during the study: one in YA-VXT + 0.5 mg CMS, two in OA-CTRL, and one in OA-VXT + 0.5 mg CMS. All SAEs were deemed unrelated to the study vaccine. The participant in OA-VXT + 0.5 mg CMS discontinued the study before Day 180 due to this SAE (invasive ductal breast carcinoma). One participant in YA-VXT + 0.5 mg CMS reported a generalized tonic-clonic seizure, an AESI, and one in OA-CTRL reported a mild autoimmune hypothyroidism, a pIMD. Both events were considered unrelated to the study vaccine.

Changes in actual values or deviations from baseline in any haematology or biochemistry clinical laboratory parameter within 28 days post-vaccination were comparable across cohorts and followed expected trends observed after vaccination (Appendix A).

### 3.4. Hemagglutination Inhibition Titers

Table 3 shows the geometric mean (95% CI) HI titers and geometric mean ratios (95% CI) against the four vaccine strains at all timepoints.

At baseline, low HI titers against all strains were observed across all cohorts. For all strains in VaxigripTetra (A/Darwin (H3N2), A/Victoria (H1N1), B/Austria, and B/Phuket), HI GMTs generally increased from baseline to Day 7, peaked at Day 28, and declined by Day 180, although they remained above baseline (Figure 3A).

In YAs, the highest HI GMTs across all strains were observed in participants receiving VXT + 0.5 mg CMS at Day 28. In OAs, the highest HI GMTs were observed in participants receiving VXT + 1 mg CMS at Day 28 for influenza A strains, and for participants receiving VXT + 0.5 mg CMS at Day 28 for influenza B strains. OAs receiving 1 mg CMS (OA-VXT + 1 mg CMS) consistently had higher HI titers against both influenza A strains at Days 7, 28, and 180 compared to OA-CTRL and OA-VXT + 0.5 mg CMS, indicating LVA’s capacity to enhance immunogenicity in OAs (Figure 3B).

HI titers against H1N1 in OA-VXT + 0.5 mg CMS were significantly lower on Days 7, 28, and 180 (*p* = 0.040, *p* = 0.015, *p* = 0.015, respectively) compared to YA-VXT + 0.5 mg CMS. Additionally, HI titers against H1N1 in OA-VXT + 0.5 mg CMS were also significantly lower compared to YA-VXT + 1 mg CMS on Days 7 and 28 (*p* = 0.043, *p* = 0.045, respectively). OA-VXT + 1 mg CMS showed significantly higher HI titers against H1N1 than OA-VXT + 0.5 mg CMS on Days 28 and 180 (*p* = 0.015, *p* = 0.036, respectively). Similar trends were observed for H3N2, though only OA-CTRL had significantly lower titers on Day 7 compared to YA-VXT + 1 mg CMS (*p* = 0.045) and on Day 180 compared to YA-VXT + 0.5 mg CMS and YA-VXT + 1 mg CMS (*p* = 0.045, *p* = 0.023, respectively).

For B/Austria, immunogenicity was generally lower than for influenza A strains, but OAs receiving LVA containing 0.5 mg CMS (OA-VXT + 0.5 mg CMS) had consistently higher HI titers on Days 7, 28, and 180 compared to YA cohorts. HI titers against B/Austria were significantly higher in OA-VXT + 0.5 mg CMS compared to YA-CTRL on Day 7 (*p* = 0.048), to YA-VXT + 1 mg CMS on Day 28 (0.040), and to all YA cohorts on Day 180 (*p* = 0.045, *p* = 0.030, *p* = 0.022 for YA-CTRL, YA-VXT + 0.5 mg CMS, YA-VXT + 1 mg CMS, respectively). Conversely, responses to B/Phuket were significantly lower in OAs than YAs across all visits, although a dose response with increased titers is visible for the LVA groups.

Using linear mixed-effects models, the strongest effect was driven by visit, with results aligning with univariate analyses (Appendix A). Residual diagnostics for HI models showed no significant deviations for H3N2, H1N1, and B/Phuket, while the B/Austria model exhibited a significant deviation from uniformity (KS test *p* = 0.014); however, no issues with outliers or dispersion were detected, supporting the overall validity of all models (Appendix A). Across all strains, HI titers increased significantly at Days 7, 28, and 180 relative to baseline (*p* < 0.001 for all strains and visits). For H3N2, significant interactions indicated a stronger early response in YA-VXT + 0.5 mg CMS (Day 7: *p* = 0.007; Day 28: *p* = 0.026), and in OA-VXT + 1 mg CMS (Days 7 and 28: *p* = 0.017 and *p* = 0.016). For H1N1, OA-CTRL and OA-VXT + 0.5 mg CMS had significantly lower baseline titers than YA-CTRL (OA-CTRL: *p* = 0.020; OA-VXT + 0.5 mg CMS: *p* = 0.023). For H1N1, there was also a significant interaction for YA-VXT + 1 mg CMS and Day 7 (*p* = 0.008). For B/Austria, only the effect of visit was significant across all post-vaccination visits (*p* < 0.001), with no cohort or interaction effects, although OA-VXT + 0.5 mg CMS approached significance at Day 7 (*p* = 0.105). For B/Phuket, baseline titers in OA-CTRL were lower (*p* = 0.045), and there were significant interactions for YA-VXT + 1 mg CMS and OA-VXT + 1 mg CMS at Day 7 (*p* = 0.028 and *p* = 0.049).

### 3.5. Seroprotection and Seroconversion

Seroprotection rates (Figure 4A) were highest for H1N1 and H3N2, with ≥81.9% of participants achieving protective HI titers at Day 28 across cohorts. These rates remained ≥60% at Day 180 in all groups. In contrast, seroprotection for the B/Austria and B/Phuket strains was lower and more variable. For B/Austria, most cohorts approached or exceeded 60% seroprotection at Day 28 but dropped in some OA groups by Day 180. For B/Phuket, rates remained consistently lower, especially in OAs, with some cohorts falling below 50% even at peak response.

When comparing age groups, YAs had higher seroprotection rates than OAs for H1N1, H3N2, and B/Phuket. However, for B/Austria, OAs in the LVA groups (particularly LVA containing 0.5 mg CMS) achieved similar or better seroprotection than YAs.

Seroconversion rates (Figure 4B) revealed a different pattern. For H1N1 and H3N2, OAs receiving vaccines with LVA showed higher seroconversion at Day 28 than YAs, indicating a robust adjuvant effect in this population. This advantage was not maintained at Day 180. For B/Austria, seroconversion was moderate in all groups, but again higher in LVA-adjuvanted OA cohorts. For B/Phuket, the highest seroconversion was observed in YAs receiving 1 mg CMS (91.7% at Day 28), while OA responses were much lower, especially in the 1 mg group (31.3%).

### 3.6. Microneutralization Titers

Appendix A shows the geometric mean (95% CI) MN titers and geometric mean ratios (95% CI) against the four vaccine strains at all timepoints. MN GMTs followed similar patterns to HI titers, generally increasing from baseline to Day 7, peaking at Day 28, and declining by Day 180—although remaining higher than baseline (Figure 5A). In both YAs and OAs, cohorts receiving LVA containing 1 mg CMS (YA-VXT + 1 mg CMS and OA-VXT + 1 mg CMS) showed the highest MN titers for H1N1 and B/Phuket on Day 28. Conversely, cohorts receiving 0.5 mg CMS (YA-VXT + 0.5 mg CMS and OA-VXT + 0.5 mg CMS) showed the highest MN titers for H3N2 and B/Austria.

The effect of LVA containing 1 mg CMS in OAs (OA-VXT + 1 mg CMS) on MN titers was less pronounced than observed for HI titers (Figure 5B). No significant differences in MN titers against H1N1 and B/Austria were detected after adjustment between any cohorts or visits. For H3N2, YA-VXT + 0.5 mg CMS and YA-VXT + 1 mg CMS, had significantly higher MN titers on Day 7 compared to YA-CTRL (*p* = 0.030 and *p* = 0.034, respectively), while OA-CTRL showed significantly lower MN titers than YA-VXT + 0.5 mg CMS and YA-VXT + 1 mg CMS (*p* = 0.030 for both). On Day 180, only OA-CTRL had significantly lower MN titers than YA-VXT + 0.5 mg CMS (*p* = 0.030). Additionally, OA-CTRL had significantly lower MN titers for B/Phuket across all visits compared to YA-VXT + 0.5 mg CMS and YA-VXT + 1 mg CMS, indicating that LVA can boost MN titers in OAs at least to the level observed in YAs. Model-based analyses of MN titers yielded similar results to the univariate analyses, with the strongest effect attributable to visit (Appendix A). Again, residual diagnostics showed no significant deviations for H3N2, H1N1, and B/Phuket, while the B/Austria model exhibited a significant deviation from uniformity (KS test *p* = 0.012), without outliers or dispersion, supporting the overall validity of models (Appendix A). Significant interactions were noted, including increased MN titers for OA-VXT + 1 mg CMS on Day 28 against H3N2 (*p* = 0.044) and B/Austria (*p* = 0.025) and for YA-VXT + 1 mg CMS on Day 7 against B/Phuket (*p* = 0.028).

## 4. Discussion

Our findings demonstrate that LVA, a novel carbohydrate fatty-acid monosulphate ester, squalane-based adjuvant, is safe and enhances the immunogenicity of a licensed quadrivalent inactivated influenza vaccine in OAs. Reactogenicity was observed with a dose–response relationship in YAs but was less pronounced in OAs. Most AEs were mild to moderate and resolved spontaneously within three days. Notably, LVA containing 1 mg of CMS significantly improved humoral responses to influenza A strains in OAs, achieving titers comparable to YAs. For B/Austria, LVA with a lower dose of 0.5 mg CMS was sufficient to elicit strong responses in OAs, whereas B/Phuket responses remained poor across all cohorts. These results support the potential of LVA containing CMS to enhance immunogenicity and mitigate the impact of immunosenescence in OAs.

These findings are consistent with previous studies demonstrating that oil-in-water-based adjuvants such as MF59 and AS03 can enhance vaccine immunogenicity in older adult populations. In individuals aged 60 years and older, both humoral (HI) and cellular (CD4+ T cells) responses were stronger in those receiving adjuvanted influenza vaccines compared to those receiving non-adjuvanted, standard-dose vaccines [18,19,20,21]. Additionally, multiple studies in this age group have shown that adjuvanted influenza vaccines (MF59 and AS03) offer improved efficacy and effectiveness over non-adjuvanted vaccines [22,23]. Therefore, adjuvantation of influenza vaccines represents a valuable strategy to improve VE in older adults and is increasingly recommended [24].

A notable finding was the differential immunogenicity elicited by the four influenza strains in the quadrivalent vaccine. While CMS-based adjuvanted formulations significantly enhanced responses to the influenza A strains (H1N1 and H3N2), and to a lesser extent B/Austria, the response to B/Phuket remained consistently low across all cohorts and visits. Variable and less robust strain-specific responsiveness have been previously reported [25,26]. Several factors contribute to this variability. First, the antigenic properties of each strain can influence how effectively the immune system recognizes and responds to it. It is hypothesized that influenza B strains elicit weaker responses due to lower immunogenicity of the HA protein or suboptimal antigen presentation. Recall responses to B strains are also known to be highly variable, especially in children [27], though this area remains understudied [25]. Second, pre-existing immunity plays a critical role; immune responses may be skewed toward previously encountered epitopes (i.e., original antigenic sin) [25]. In this study, relatively high baseline HI titers may have influenced the magnitude of the post-vaccination response. Nonetheless, the low baseline and post-vaccination titers for B/Phuket suggest a lack of immunological priming. Third, host factors, including age-related immune remodelling, can differentially affect responses to specific strains. Older adults often exhibit diminished B-cell diversity and T-cell help, which may disproportionately impact responses to less immunogenic strains like B/Phuket [7,25], although low immunogenicity for B/Phuket was also observed for YAs in this study.

LVA, with CMS as the active ingredient, is a novel class of synthetic adjuvants with a distinct mechanism of action. Typically, oil-in-water emulsion adjuvants enhance immune responses by generating an “immunocompetent environment” at the injection site [28]. This environment is characterized by the production of pro-inflammatory cytokines, recruitment of innate immune cells, and uptake of antigen by these cells. The antigen and adjuvant are then transported to the draining lymph node, where they activate B- and T-cells [28,29,30]. The exact mechanisms of action of LVA and CMS in exerting their adjuvant effect remain unknown. While HI titers are established correlates of protection, unravelling the mechanisms of action of CMS will provide a deeper understanding of the immunogenicity data observed.

The study employed a rigorous, randomized, observer-blind design with well-defined cohorts. Including both younger and older adults enabled direct age-stratified comparisons, and the use of a licensed quadrivalent inactivated influenza vaccine (VaxigripTetra) as the control permitted direct assessment of the adjuvant’s effects at two dose levels. However, the relatively small sample size, although typical for phase 1 studies, limited the study’s power to detect rare adverse events or subtle differences in immunogenicity, and the single-centre design may affect the generalizability of the findings. Another limitation relates to the identification of intercurrent influenza infections during the study. While symptomatic cases were monitored by antigen self-testing, this approach is less sensitive than PCR. We did not observe unexpectedly high antibody titers at any timepoint, suggesting that unrecognized infections, if any, were rare and unlikely to have affected the results. Additionally, the consistently poor response to B/Phuket across all groups warrants further investigation. The capacity of LVA to enhance responses to influenza A strains and B/Austria, but not B/Phuket, suggests that while the adjuvant is effective, its impact may be influenced by the intrinsic properties of the antigen or by host factors. Notably, B/Phuket (Yamagata lineage) was not included in the 2024–2025 vaccine because B/Yamagata has not been observed in seasonal epidemics since 2020. Although immunogenicity was thoroughly assessed via HI and MN assays, earlier data from a phase 1a trial in younger adults also demonstrated the ability of LVA to induce influenza-specific CD4^+^ T-cell responses. These findings suggest that the immunostimulatory effects of LVA extend beyond humoral immunity. Future studies should further elucidate its mechanisms of action and potential to enhance cell-mediated responses in older adults.

## 5. Conclusions

In conclusion, LVA is a promising adjuvant capable of enhancing influenza vaccine immunogenicity in OAs, particularly against influenza A strains and B/Austria (Victoria lineage). LVA containing a dose of 1 mg CMS appears optimal for further development, as it offers robust immunogenicity without a marked increase in reactogenicity in OAs. Further research is needed to optimize vaccine formulations and elucidate the mechanisms of action. Overall, these findings support the continued clinical development of CMS-based adjuvanted influenza vaccines, especially to protect vulnerable older populations.

## Figures and Tables

**Figure 1 vaccines-13-00922-f001:**
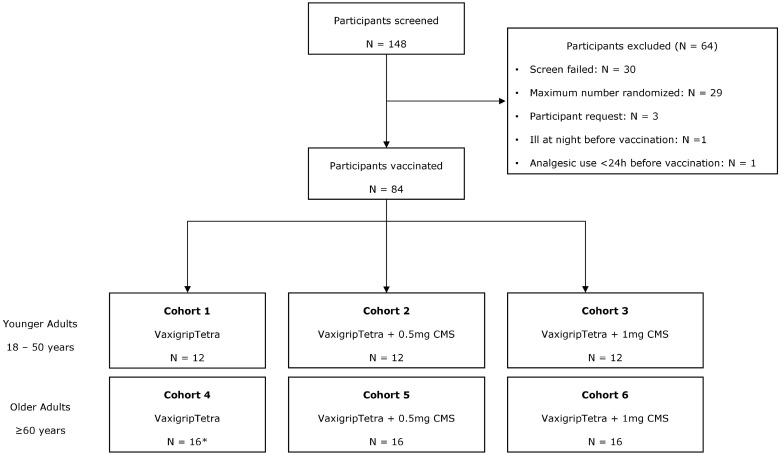
CONSORT flowchart summarizing the participant flow, including participant recruitment, inclusion, and randomization. * One participant was excluded from the IAS because of existing undiagnosed hypothyroidism.

**Figure 2 vaccines-13-00922-f002:**
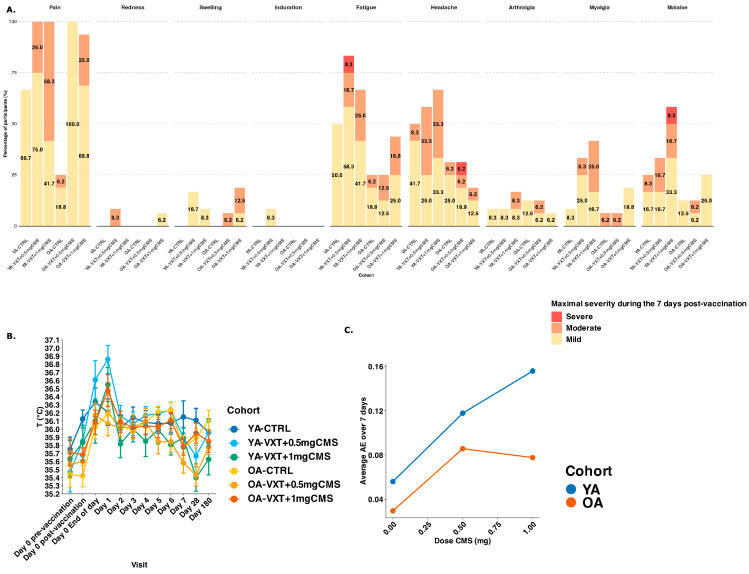
Summary of solicited Adverse Events. (**A**) Number of participants per cohort reporting local or systemic adverse events for at least one day of mild, moderate, or severe intensity (maximum intensity over the 7-day reporting period shown). (**B**) Oral body temperature (in °C) at baseline (pre-vaccination) and for each timepoint during safety follow-up. (**C**) Average AE severity across 7 days for all YAs and OAs shown per dose of CMS in mg. YA = younger adults aged 18–50 years; OA = older adults 60 years or older. YA-CTRL = YA, VaxigripTetra; YA-VXT + 0.5 mg CMS = YA, VaxigripTetra + LVA containing 0.5 mg CMS; YA-VXT + 1 mg CMS = YA, VaxigripTetra + LVA containing 1 mg CMS; OA-CTRL = OA, VaxigripTetra; OA-VXT + 0.5 mg CMS = OA, VaxigripTetra + LVA containing 0.5 mg CMS; OA-VXT + 1 mg CMS = OA, VaxigripTetra + LVA containing 1 mg CMS.

**Figure 3 vaccines-13-00922-f003:**
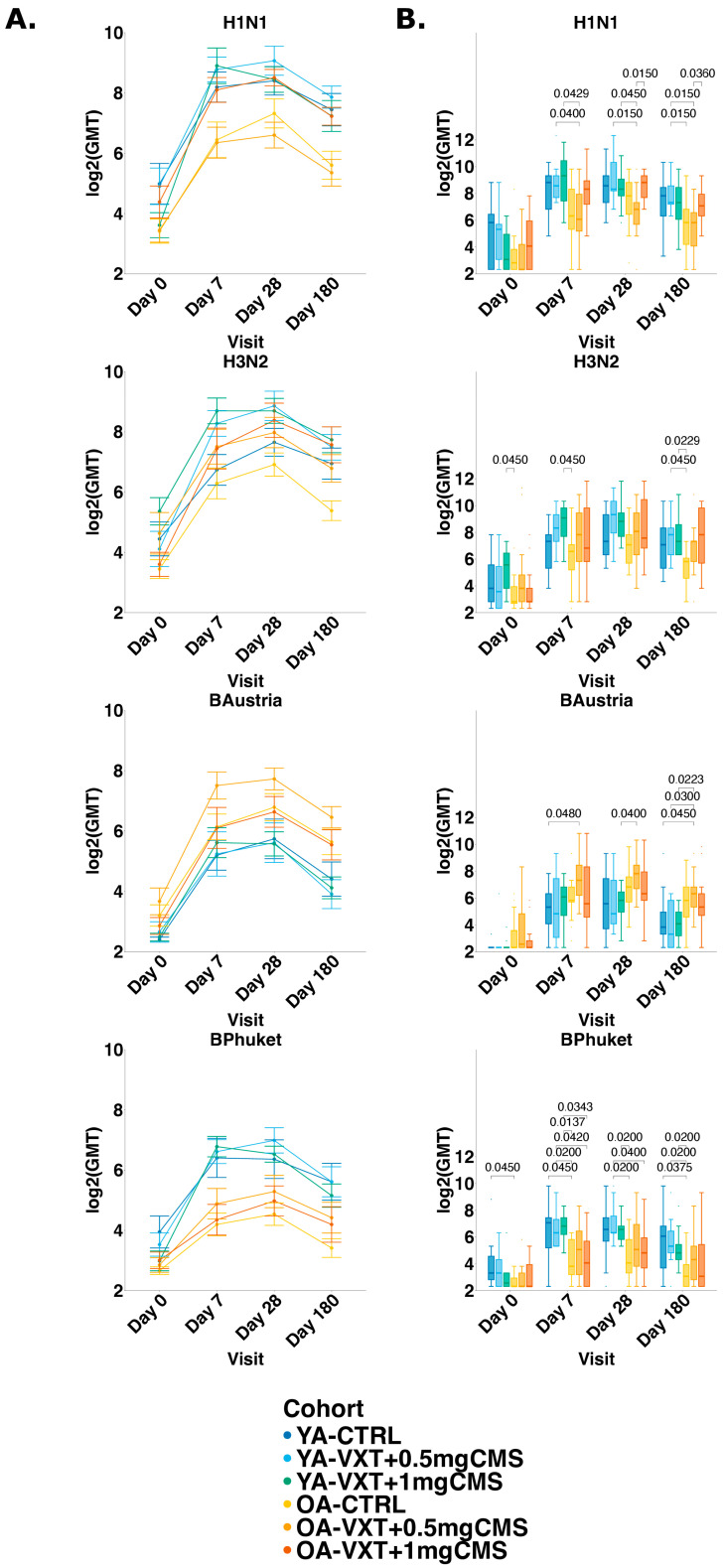
Summary of Haemagglutination inhibition titers. (**A**) Geometric mean of log2-transformed titers per cohort for each vaccine strain on all study visits. Dots represent GMT with SE bars. (**B**) Boxplots showing median (IQR) of log2-transformed HI titers per cohort for each vaccine strain on all study visits. *p*-values (adjusted) between cohorts at each visit are calculated using Wilcoxon rank-sum test with Benjamini–Hochberg procedure for correction of multiple testing. Only significant adjusted *p*-values are shown. YA = younger adults aged 18–50 years; OA = older adults 60 years or older. YA-CTRL = YA, VaxigripTetra; YA-VXT + 0.5 mg CMS = YA, VaxigripTetra + LVA containing 0.5 mg CMS; YA-VXT + 1 mg CMS = YA, VaxigripTetra + LVA containing 1 mg CMS; OA-CTRL = OA, VaxigripTetra; OA-VXT + 0.5 mg CMS = OA, VaxigripTetra + LVA containing 0.5 mg CMS; OA-VXT + 1 mg CMS = OA, VaxigripTetra + LVA containing 1 mg CMS.

**Figure 4 vaccines-13-00922-f004:**
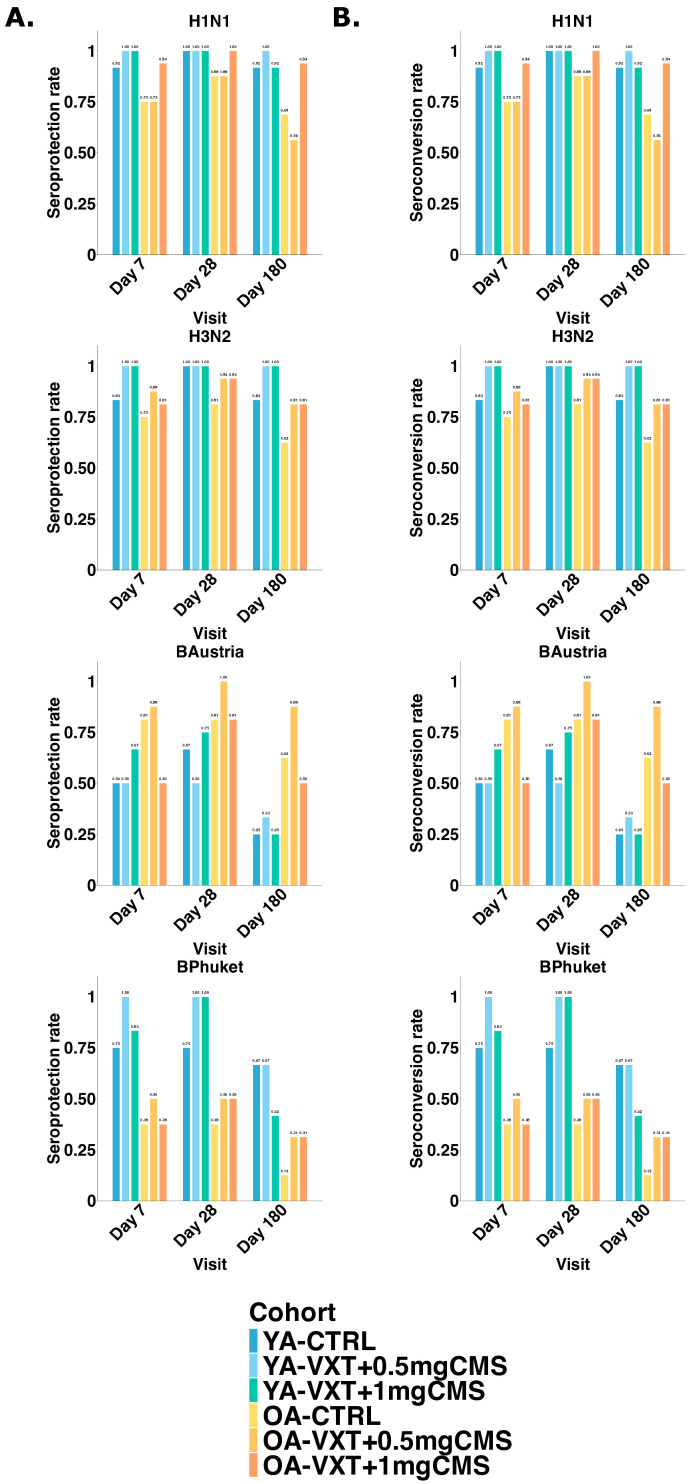
Seroprotection and seroconversion rates based on HI titers. (**A**). Seroprotection rates per cohort for each vaccine strain on all study visits. Seroprotection was defined as participants with an HI antibody titer ≥ 40. (**B**) Seroconversion rates per cohort for each vaccine strain on all study visits. Seroconversion was characterized by a 4-fold increase in HI antibody titer compared to pre-vaccination level. YA = younger adults aged 18–50 years; OA = older adults 60 years or older. YA-CTRL = YA, VaxigripTetra; YA-VXT + 0.5 mg CMS = YA, VaxigripTetra + LVA containing 0.5 mg CMS; YA-VXT + 1 mg CMS = YA, VaxigripTetra + LVA containing 1 mg CMS; OA-CTRL = OA, VaxigripTetra; OA-VXT + 0.5 mg CMS = OA, VaxigripTetra + LVA containing 0.5 mg CMS; OA-VXT + 1 mg CMS = OA, VaxigripTetra + LVA containing 1 mg CMS.

**Figure 5 vaccines-13-00922-f005:**
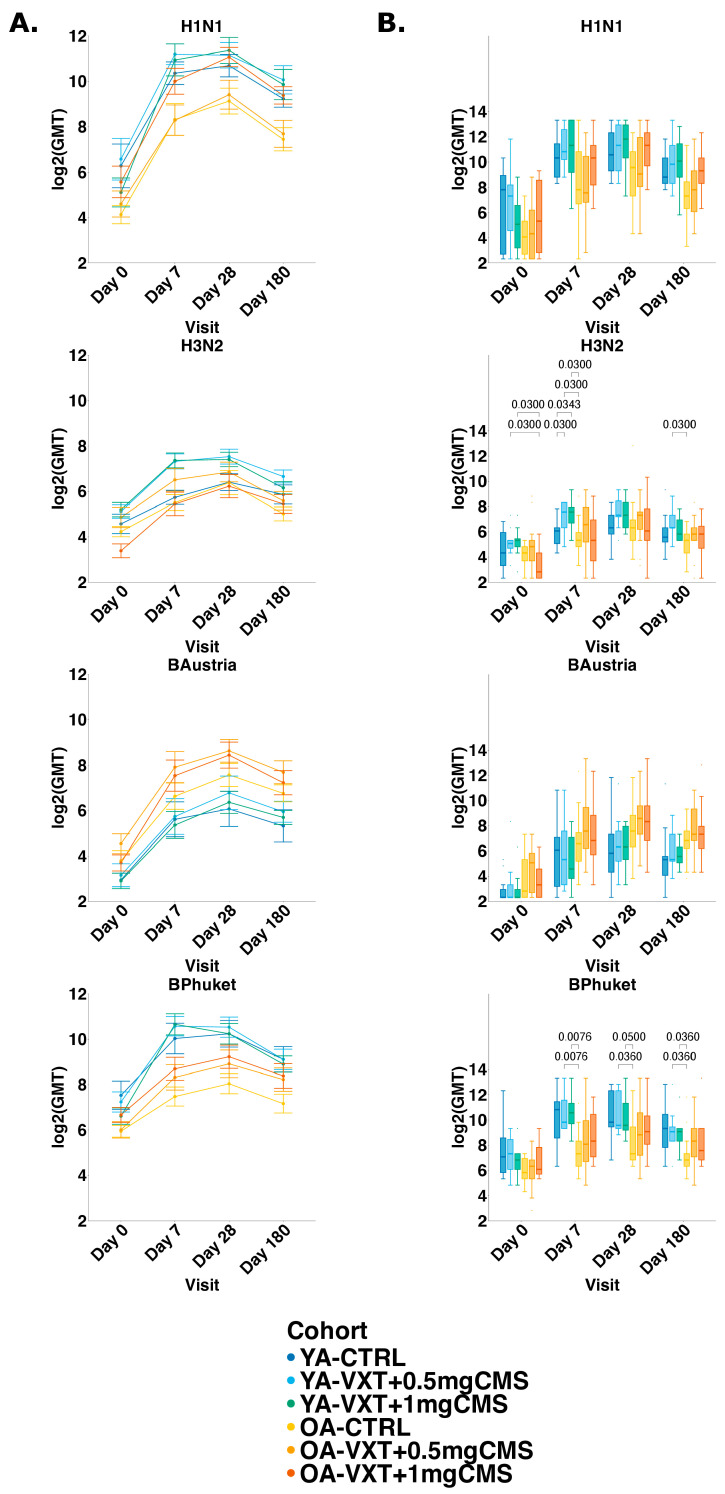
Summary of microneutralization titers. (**A**) Geometric mean of log2-transformed titers per cohort for each vaccine strain on all study visits. Dots represent GMT with SE bars. (**B**) Boxplots showing median (IQR) of log2-transformed HI titers per cohort for each vaccine strain on all study visits. *p*-values (adjusted) between cohorts at each visit are calculated using Wilcoxon rank-sum test with Benjamini–Hochberg procedure for correction of multiple testing. Only significant adjusted *p*-values are shown. YA = younger adults aged 18–50 years; OA = older adults 60 years or older. YA-CTRL = YA, VaxigripTetra; YA-VXT + 0.5 mg CMS = YA, VaxigripTetra + LVA containing 0.5 mg CMS; YA-VXT + 1 mg CMS = YA, VaxigripTetra + LVA containing 1 mg CMS; OA-CTRL = OA, VaxigripTetra; OA-VXT + 0.5 mg CMS = OA, VaxigripTetra + LVA containing 0.5 mg CMS; OA-VXT + 1 mg CMS = OA, VaxigripTetra + LVA containing 1 mg CMS.

**Table 1 vaccines-13-00922-t001:** Demographic characteristics of randomized and vaccinated participants in each cohort.

		YA-CTRL (N = 12)	YA-VXT + 0.5 mg CMS (N = 12)	YA-VXT + 1 mg CMS (N = 12)	OA-CTRL (N = 16)	OA-VXT + 0.5 mg CMS (N = 16)	OA-VXT + 1 mg CMS (N = 16)	Total (N = 84)
Age (years (SD))		28.9 (9.0)	28.3 (8.6)	27.8 (7.7)	63.6 (2.6)	66.9 (6.2)	67.2 (5.0)	49.8 (19.8)
Gender (n, %)	Female	8 (66.7%)	7 (58.3%)	8 (66.7%)	8 (50.0%)	8 (50.0%)	9 (56.3%)	48 (57.1%)
	Male	4 (33.3%)	5 (41.7%)	4 (33.3%)	8 (50.0%)	8 (50.0%)	7 (43.8%)	36 (42.9%)
Race (n,%)	White	12 (100.0%)	12 (100.0%)	10 (83.3%)	16 (100.0%)	16 (100.0%)	15 (93.8%)	81 (96.4%)
	Asian	0 (0.0%)	0 (0.0%)	2 (16.7%)	0 (0.0%)	0 (0.0%)	1 (6.3%)	3 (3.6%)
Weight (kg, (SD))		72.37 (10.19)	71.99 (13.21)	66.51 (10.21)	77.55 (11.47)	74.38 (7.92)	75.04 (15.60)	73.36 (11.88)
BMI (kg/m^2^ (SD))		24.90 (3.26)	25.76 (4.69)	22.23 (2.69)	24.96 (2.54)	25.14 (2.24)	26.24 (3.82)	24.95 (3.39)

BMI = body mass index, SD = standard deviation. % was based on only female participants. YA = younger adults aged 18–50 years; OA = older adults 60 years or older. YA-CTRL = YA, VaxigripTetra; YA-VXT + 0.5 mg CMS = YA, VaxigripTetra + LVA containing 0.5 mg CMS; YA-VXT + 1 mg CMS = YA, VaxigripTetra + LVA containing 1 mg CMS; OA-CTRL = OA, VaxigripTetra; OA-VXT + 0.5 mg CMS = OA, VaxigripTetra + LVA containing 0.5 mg CMS; OA-VXT + 1 mg CMS = OA, VaxigripTetra + LVA containing 1 mg CMS.

**Table 2 vaccines-13-00922-t002:** Summary of unsolicited adverse events in each cohort.

	YA-CTRL (N = 12)	YA-VXT + 0.5 mg CMS (N = 12)	YA-VXT + 1 mg CMS (N = 12)	OA-CTRL (N = 16)	OA-VXT + 0.5 mg CMS (N = 16)	OA-VXT + 1 mg CMS (N = 16)	Total (N = 84)
Any (nP, %, nE)	8 (66.7%) 12	11 (91.7%) 20	11 (91.7%) 39	11 (68.8%) 20	11 (68.8%) 16	13 (81.3%) 29	65 (77.4%) 136
Any severe AEs (Grade 3 or higher, (nP, %, nE))	1 (8.3%) 2	0 (0.0%) 0	1 (8.3%) 2	0 (0.0%) 0	2 (12.5%) 2	1 (6.3%) 1	5 (6.0%) 7
Probably related (nP, %, nE)	1 (8.3%) 1	4 (33.3%) 5	6 (50.0%) 7	2 (12.5%) 3	0 (0.0%) 0	4 (25.0%) 7	17 (20.2%) 23
Definitely related (nP, %, nE)	1 (8.3%) 1	5 (41.7%) 7	10 (83.3%) 22	1 (6.3%) 1	1 (6.3%) 1	8 (50.0%) 10	26 (31.0%) 42
Severe and related * AEs (nP, %, nE)	0 (0.0%) 0	0 (0.0%) 0	1 (8.3%) 2	0 (0.0%) 0	0 (0.0%) 0	0 (0.0%) 0	1 (1.2%) 2

AE = adverse event, nE = number of events, nP = number of participants with at least 1 event. * Related is defined here as “probably related” or “definitely related”. YA = younger adults aged 18–50 years; OA = older adults 60 years or older. YA-CTRL = YA, VaxigripTetra; YA-VXT + 0.5 mg CMS = YA, VaxigripTetra + LVA containing 0.5 mg CMS; YA-VXT + 1 mg CMS = YA, VaxigripTetra + LVA containing 1 mg CMS; OA-CTRL = OA, VaxigripTetra; OA-VXT + 0.5 mg CMS = OA, VaxigripTetra + LVA containing 0.5 mg CMS; OA-VXT + 1 mg CMS = OA, VaxigripTetra + LVA containing 1 mg CMS.

**Table 3 vaccines-13-00922-t003:** Geometric mean haemagglutination inhibition (HI) antibody titers (GMTs), per cohort per vaccine strain at baseline and 7, 28, and 180 days after vaccination.

			H3N2			H1N1			B/Austria			B/Phuket		
YA	Visit	Statistic	CTRL	VXT + 0.5 mg CMS	VXT + 1 mg CMS	CTRL	VXT + 0.5 mg CMS	VXT + 1 mg CMS	CTRL	VXT + 0.5 mg CMS	VXT + 1 mg CMS	CTRL	VXT + 0.5 mg CMS	VXT + 1 mg CMS
	Day 0	GMT	21.8	17.3	41.2	31.7	30.0	12.2	5.3	6.3	5.6	15.4	11.6	7.9
		95% CI	9.3–51.3	7.1–42.3	20.6–82.3	11.3–88.9	11.9–75.5	6.5–23.1	4.7–6.0	3.8–10.5	4.6–6.8	6.9–34.6	6.4–20.8	4.8–13.1
	Day 7	GMT	106.8	310.9	415.0	293.4	439.7	479.5	36.7	37.8	49.0	84.8	97.9	109.9
		95% CI	49.0–232.7	161.7–597.7	216.3–796.0	136.7–629.7	235.1–822.3	195.0–1179.0	17.2–78.3	12.3–116.0	22.9–104.7	31.5–227.8	53.5–179.2	65.5–184.5
		GMR	4.9	18.0	10.1	9.2	14.7	39.2	6.9	6.0	8.7	5.5	8.5	13.8
		95% CI	2.7–8.9	6.0–53.5	3.9–25.9	2.5–34.0	4.3–49.8	11.6–132.5	3.3–14.5	2.0–18.1	4.4–17.3	2.0–15.1	3.3–21.8	6.5–29.4
	Day 28	GMT	201.6	465.8	415.0	339.0	538.2	349.0	53.4	49.0	47.6	82.3	127.0	92.4
		95% CI	99.6–407.9	221.8–978.3	220.9–779.8	165.0–696.6	257.7–1123.9	183.2–664.8	19.6–145.4	18.0–133.1	25.8–87.7	31.0–218.9	67.0–240.6	61.7–138.4
		GMR	9.2	26.9	10.1	10.7	18.0	28.5	10.1	7.8	8.5	5.3	11.0	11.6
		95% CI	4.1–21.0	8.2–88.7	4.5–22.4	3.1–37.1	4.0–80.5	11.0–73.7	3.8–26.7	2.7–22.1	4.7–15.3	2.2–13.1	3.9–30.8	6.0–22.5
	Day 180	GMT	123.4	179.6	213.6	174.5	232.9	151.0	21.2	15.0	17.3	49.0	49.0	35.6
		95% CI	56.2–270.8	94.1–342.6	110.8–411.6	76.9–395.9	133.2–407.4	68.9–331.0	8.9–50.3	7.2–31.2	10.0–30.0	19.3–124.3	22.9–104.7	19.8–64.2
		GMR	5.7	10.4	5.2	5.5	7.8	12.3	4.0	2.4	3.1	3.2	4.2	4.5
		95% CI	2.5–12.6	3.7–29.0	2.9–9.3	1.9–16.3	2.3–26.7	5.1–29.6	1.8–9.1	1.2–4.9	1.8–5.3	1.7–6.1	1.6–11.5	2.3–8.7
**OA**	**Visit**	**Statistic**	**CTRL**	**VXT +** **0.5 mg CMS**	**VXT +** **1 mg CMS**	**CTRL**	**VXT +** **0.5 mg CMS**	**VXT +** **1 mg CMS**	**CTRL**	**VXT +** **0.5 mg CMS**	**VXT +** **1 mg CMS**	**CTRL**	**VXT +** **0.5 mg CMS**	**VXT +** **1 mg CMS**
	Day 0	GMT	11.2	24.8	12.2	10.5	10.9	20.9	9.5	12.7	7.2	6.2	7.2	8.1
		95% CI	6.9–18.3	9.0–68.6	6.7–22.0	5.6–19.7	6.1–19.6	9.5–45.7	5.5–16.7	6.6–24.5	4.8–10.9	5.1–7.4	5.0–10.4	5.4–12.1
	Day 7	GMT	83.8	182.2	174.5	89.8	81.8	275.0	81.9	182.2	68.7	18.2	29.5	20.4
		95% CI	37.7–186.0	77.7–427.4	64.1–474.9	35.2–229.1	38.3–174.5	149.9–504.5	45.7–146.6	93.8–353.8	25.2–187.3	10.0–33.1	14.0–62.4	9.6–43.4
		GMR	7.5	7.3	14.4	8.6	7.5	13.2	8.6	14.4	9.5	3.0	4.1	2.5
		95% CI	3.5–16.0	2.7–19.7	6.3–32.5	4.0–18.6	3.6–15.5	5.5–31.4	4.6–16.0	5.7–36.4	3.7–24.6	1.7–5.1	2.1–8.0	1.2–5.2
	Day 28	GMT	121.3	252.2	334.2	145.9	97.2	364.4	127.0	212.0	99.3	24.1	39.1	31.5
		95% CI	65.7–223.6	119.8–530.6	144.4–773.4	70.0–304.0	51.7–183.0	243.3–545.8	67.0–240.7	124.0–362.5	46.8–211.1	13.4–43.3	17.8–86.3	15.1–65.9
		GMR	10.8	10.2	27.5	13.9	8.9	17.4	13.3	16.7	13.7	3.9	5.4	3.9
		95% CI	5.9–19.8	4.0–25.7	13.7–55.1	7.1–27.4	4.7–16.8	8.1–37.4	6.1–28.8	7.6–36.9	7.0–27.1	2.2–6.8	2.7–10.9	1.9–7.9
	Day 180	GMT	40.9	110.6	190.3	40.9	40.9	149.9	54.0	87.7	46.5	10.7	21.4	18.3
		95% CI	24.5–68.5	56.1–217.9	78.8–459.6	21.9–76.6	21.2–79.1	96.7–232.4	28.5–102.4	51.9–148.4	22.2–97.4	6.5–17.5	10.0–45.7	7.7–43.8
		GMR	3.6	6.1	15.7	3.9	3.6	7.2	5.7	6.5	6.4	1.7	2.9	2.3
		95% CI	2.4–5.7	3.2–11.7	7.3–33.7	2.0–7.7	2.1–6.0	3.5–14.6	2.7–11.6	2.9–14.6	3.6–11.5	1.1–2.7	1.6–5.3	1.1–4.8

GMT: geometric mean titer; GMR: geometric mean ratio compared to Day 0; 95% CI: 95% confidence interval. YA = younger adults aged 18–50 years; OA = older adults 60 years or older; CTRL = VaxigripTetra; VXT + 0.5 mg CMS = VaxigripTetra + LVA containing 0.5 mg CMS; VXT + 1 mg CMS = VaxigripTetra + LVA containing 1 mg CMS.

## Data Availability

Raw data, including de-identified participant data, will be made available upon reasonable request to the corresponding author.

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
