# Peer review of "A Novel Carbohydrate Fatty-Acid Monosulphate Ester, Squalane-in-Water Adjuvant Is Safe and Enhances Inactivated Influenza Vaccine Immunogenicity in Older Adults"

_vaccines, 2025, doi:10.3390/vaccines13090922_

Round 1
Reviewer 1 Report
Comments and Suggestions for Authors
This manuscript reports a Phase 1 clinical trial of the use of a novel adjuvant in combination with a clinically approved quadrivalent influenza vaccine in two adult age groups. Group 1 age 18 – 50 years. Group 2 age >60 years. The vaccine alone is used as baseline control, and 2 concentrations of adjuvant are tested.
The same group have published characterisation of this adjuvant previously.
Measured outcomes are adverse reactions and HI titres for the influenza strains covered by the vaccine.
The conclusions drawn are.
- The use of the adjuvant is safe in both groups and at both tested concentrations. Displaying an acceptable adverse reaction profile.
- The adjuvant significantly increases response (HI titres) to both strains of influenza A and 1 of the 2 strains of influenza B in both age groups and for both concentrations. The possible reasons for this are well discussed.
The experiments are well designed and carried out.
These conclusions are supported by the data presented.
However, significant improvements to the resolution of the graphical data in figures 2, 3, 4, 5 and S2 is required to ensure that these will be legible in the published version. Specifically, the font sizes need to be increased for all labelling and the colours changed to give clearer contrasts between data sets.
Author Response
We thank the reviewer for their helpful comment. We agree that the resolution of the figures in the submitted version was not optimal, as they had been placed in the body of the text, and subsequently compressed into a single PDF file. The figures have now been uploaded as separate high-resolution PDF files. Additionally, we have improved the graphical quality of all figures to have clearer lines, dots and boxes, as well as labels in increased font size. This also makes the contrast between colours better: Blue-Turquoise-Green for YA cohorts and Yellow-Orange-Red for OA cohorts.
Reviewer 2 Report
Comments and Suggestions for Authors
This manuscript presents the results of a clinical trial of a new adjuvant designed to enhance the immunogenicity of inactivated influenza vaccines. The safety and tolerability of this adjuvant were studied in two age cohorts - young and older adults, when added to a commercial 4-valent inactivated vaccine. The immunogenicity of the vaccine in the presence of the new adjuvant was assessed in comparison with the vaccine without adjuvant. The data obtained are valuable and can be used for further implementation of the new adjuvant composition. The article is written quite well, but there are a number of comments.
- Table 2 should be moved to the Results section, as it reports the results of safety study of the tested vaccines.
- Please edit the Figures 2-5 so that the labels on the axes and the values of the boxplots are distinguishable.
- Figure 3 shows the HI titers in serum samples against all four influenza strains, but it is unclear which absolute numerical titer values are shown. The text also simply states that the GMT in one group is greater than the other, but the numerical values of the titres are not given. Figure legend states that there are 2log-transformed values, but Y axis is shown as log10(GMT). It is difficult to imagine that the HI titer can be 300 log10 (e.g. 10 to the power of 300).
Author Response
This manuscript presents the results of a clinical trial of a new adjuvant designed to enhance the immunogenicity of inactivated influenza vaccines. The safety and tolerability of this adjuvant were studied in two age cohorts - young and older adults, when added to a commercial 4-valent inactivated vaccine. The immunogenicity of the vaccine in the presence of the new adjuvant was assessed in comparison with the vaccine without adjuvant. The data obtained are valuable and can be used for further implementation of the new adjuvant composition. The article is written quite well, but there are a number of comments.
- Table 2 should be moved to the Results section, as it reports the results of safety study of the tested vaccines.
- Please edit the Figures 2-5 so that the labels on the axes and the values of the boxplots are distinguishable.
- Figure 3 shows the HI titers in serum samples against all four influenza strains, but it is unclear which absolute numerical titer values are shown. The text also simply states that the GMT in one group is greater than the other, but the numerical values of the titres are not given. Figure legend states that there are 2log-transformed values, but Y axis is shown as log10(GMT). It is difficult to imagine that the HI titer can be 300 log10 (e.g. 10 to the power of 300).
We thank the reviewer for their constructive feedback, which has helped us to improve the clarity and quality of the manuscript. Regarding their questions, please find the answers:
- Table 2 was positioned at the end of the manuscript file but is mentioned in Line 243 (TC version), in the Safety paragraph of the Results section. It is indeed an important table that needs reporting in the Results section.
- We agree that the quality of the figures in the submitted version was suboptimal, as they had been placed in the body of the text, and subsequently compressed into a single PDF file. The figures have now been uploaded as separate high-resolution PDF files. Additionally, we have improved the graphical quality of all figures to have clearer lines, dots and boxes, as well as labels in increased font size.
- The numerical variables of the HI titres were provided in Supplementary Table 3. We agree with the reviewer that they are important for the interpretation of the figures, and the table has now been moved to the main text as Table 3. The Y-axis of Figure 3 was indeed transformed and labelled incorrectly. It does concern 2log-transformed values not log10. This has now been updated.
Reviewer 3 Report
Comments and Suggestions for Authors
Why was chosen the inactivated vaccine to evaluate the adjuvant safety profile and immunogenicity of the new adjuvant, being the recombinant vaccine less immunogenic, that would benefit from a potent adjuvant, would be the more interesting choice?
The new adjuvant is based on CMS combined with a squalane-in-water emulsion, authors in discussion compare with oil-in-water-based adjuvants such as MF59 and AS03, thus it would be more interesting to compare the new adjuvant CMC alone with the CMC combined with a squalane-in-water emulsion.
Figure 2A needs to be improved to be readable!
Figures 3A, 3B, 4A, 4B, 5A and 5B need to be improved and the possible way is to separate the 4 graphics shown side by side and present them side by side only two thus the letters can be bigger and readable.
Author Response
We thank the reviewer for their insightful comment, which have helped us to further improve the manuscript, particularly the Discussion section. Our point-by-point answers are provided below:
Why was chosen the inactivated vaccine to evaluate the adjuvant safety profile and immunogenicity of the new adjuvant, being the recombinant vaccine less immunogenic, that would benefit from a potent adjuvant, would be the more interesting choice?
We chose the inactivated quadrivalent influenza vaccine because this was the standard of care recommended in Belgium at the time of the study, and recombinant influenza vaccines were (and still are) not available in Belgium. During the relevant influenza season, the Belgian NITAG advised the use of unadjuvanted inactivated influenza vaccines for all risk groups, and high-dose inactivated influenza vaccines only for older adults residing in nursing homes.
It should be noted that recombinant influenza vaccines have been shown to be more immunogenic than standard egg-based inactivated vaccines. In many countries worldwide, however, standard dose egg-based inactivated vaccines remain the backbone of seasonal influenza vaccination programs and are the most common comparator in clinical studies. For these scientific and practical reasons, the inactivated vaccine was the appropriate and most relevant comparator in our study. The same vaccine was subsequently used as the backbone for testing the addition of LVA.
The new adjuvant is based on CMS combined with a squalane-in-water emulsion, authors in discussion compare with oil-in-water-based adjuvants such as MF59 and AS03, thus it would be more interesting to compare the new adjuvant CMC alone with the CMC combined with a squalane-in-water emulsion.
LVA, which combines CMS with a squalane-in-water emulsion is the final adjuvant formulation that has undergone rigorous preclinical safety and toxicity testing and has been approved for clinical evaluation in combination with antigens. CMS alone is not approved for human use and therefore cannot be tested as a single component in a clinical trial. CMS serves as the immunostimulant component of LVA, while the squalane droplets act as the vehicle for delivery of the CMS molecules. The immune-enhancing effect of LVA is thought to result from the synergy between CMS and the emulsion. While the exact mechanisms of action of LVA are still being investigated, we agree with the reviewer that studying the effect of CMS alone would be of interest. Such experiments are currently being conducted in mouse models, but data are not available yet.
Figure 2A needs to be improved to be readable! Figures 3A, 3B, 4A, 4B, 5A and 5B need to be improved and the possible way is to separate the 4 graphics shown side by side and present them side by side only two thus the letters can be bigger and readable.
We appreciate the reviewer’s suggestion and acknowledge that the resolution of the submitted figures was suboptimal, as they had been placed directly in the manuscript text and compressed into a single PDF. We have now uploaded all figures as separate high-resolution PDF files. In addition, we have improved readability by increasing font sizes, enhancing line and marker clarity, and refining the color schemes (blue–turquoise–green for younger adults and yellow–orange–red for older adults). We believe these changes substantially improve the legibility of all figures.
Reviewer 4 Report
Comments and Suggestions for Authors
The manuscript "A novel carbohydrate fatty-acid monosulphate ester, squalane- in-water adjuvant is safe and enhances inactivated influenza vaccine immunogenicity in older adults" aims to evaluate the safety, reactogenicity, and immunogenicity of a novel CMS-based LiteVax Adjuvant with inactivated influenza vaccine in older adults, comparing responses with those in younger adults. Although the paper presents a certain scientific interest, there are some concerns regarding the validity and the overall results. Here are some important comments:
- Explain blinding procedures and distinguish between "double-blind" and "observer-blind."
- Present adjusted GMT ratios with 95% CIs and modify immunogenicity comparisons for baseline titres (ANCOVA or LME).
- For between-group comparisons, use the model-based approach or the Wilcoxon rank-sum in place of the Wilcoxon signed-rank (paired).
- Provide diagnostics and effect size estimates; fully describe covariance structure, fixed/random effects, and handling of missing data.
- Provide complete adjusted p-value tables in the Supplementary material; pre-specify the primary endpoints.
- Explain late-onset local reactions; give SAE/AESI narratives; and explain assessor blinding and causality assessment criteria.
- Future OA cohorts should incorporate measures of cellular immunity; if at all possible, correlate HI/MN with functional assays.
- To identify asymptomatic infections, consider scheduled PCR or serology; emphasize assay validation in the body of the text.
- Increase recruitment to include frail and racially/ethnically diverse OA populations.
Author Response
We thank the reviewer for their detailed and constructive feedback. These comments have helped us to clarify several methodological aspects and strengthen the manuscript.
The manuscript "A novel carbohydrate fatty-acid monosulphate ester, squalane- in-water adjuvant is safe and enhances inactivated influenza vaccine immunogenicity in older adults" aims to evaluate the safety, reactogenicity, and immunogenicity of a novel CMS-based LiteVax Adjuvant with inactivated influenza vaccine in older adults, comparing responses with those in younger adults. Although the paper presents a certain scientific interest, there are some concerns regarding the validity and the overall results. Here are some important comments:
- Explain blinding procedures and distinguish between "double-blind" and "observer-blind."
We thank the reviewer for raising this important point. The study was formally submitted and approved as an observer-blind trial. In practice, however, both participants and outcome assessors (investigators and trial nurses) were blinded: participants were not informed of their allocation and were asked to turn their head during the injection, and assessors were unaware of group assignments. Only the vaccinators (unblinded nurses) were aware of the allocation, due to visible differences in syringe appearance, but they were not involved in any safety or immunogenicity assessments. For this reason, while the protocol terminology was “observer-blind,” the conduct of the study was equivalent to double-blind with the exception of vaccinators. The Methods section has been updated (lines 129-136, TC version) to clarify these procedures.
- Present adjusted GMT ratios with 95% CIs and modify immunogenicity comparisons for baseline titres (ANCOVA or LME).
The numerical variables of the HI titres, including GMT ratios with 95% CIs were provided in Supplementary Table 3. We agree with the reviewer that they are important for the interpretation of the figures, and the table has now been moved to the main text as Table 3. Similarly, the numerical variables of the MN titres, including 95% CIs are provided in Supplementary Table 5.
- For between-group comparisons, use the model-based approach or the Wilcoxon rank-sum in place of the Wilcoxon signed-rank (paired).
We confirm that between-group comparisons were performed using the Wilcoxon rank-sum test (unpaired). The Methods section mistakenly referred to the Wilcoxon signed-rank test, which has now been corrected P-values in the Results section already reflect the correct test. The model-based approach was an additional explorative analysis that adjusts for baseline, interaction effects and random subject effects. Both approaches have been described in the Results section and yielded similar results supporting the same biological conclusions.
- Provide diagnostics and effect size estimates; fully describe covariance structure, fixed/random effects, and handling of missing data.
Model assumptions were evaluated using both visual and simulation-based diagnostics. We performed diagnostic checks using the DHARMa package in R for all linear mixed-effects models. Q-Q and residual vs. predicted plots for all models (both HI and MN models per strain) have now been included in the Supplementary information (Supplementary Figures 3 and 4).
Residuals were approximately normally distributed, with no evidence of heteroscedasticity or influential outliers. Random effects were normally distributed, and no overdispersion was detected. These findings support the validity of the linear mixed-effects models. For the B/Austria strain, residual diagnostics indicated no issues with outliers (HI: p = 0.75; MN: p = 1) or dispersion (HI and MN: p = 0.63). However, the Kolmogorov–Smirnov test revealed significant deviations from uniformity in both models (HI: p = 0.014; MN: p = 0.012), and quantile deviations were detected in the HI model’s residuals vs. predicted plot. These findings suggest potential model misspecification or residual non-normality. We have retained the models given their overall robustness and absence of influential data points but acknowledge these limitations in the Results section on Lines 303-306 and Lines 353-356 (TC version).
Effect size estimates, confidence intervals, and p-values for each predictor and interaction term are reported in Supplementary Tables 3 and 5. Geometric mean ratios (GMRs) with 95% CIs for HI and MN titres across timepoints and cohorts are reported in Table 3 and Supplementary Table 4.
The covariance structure of the models is described by the random intercepts for subjects, variance components (σ² and τ₀), intraclass correlation (ICC), and marginal/conditional R² values, all shown in Supplementary Tables 3 and 5.
Fixed effects include cohort, visit, and their interaction while Random effects are defined as subject-level intercepts, all of which are reported in Supplementary Table 3 and 5.
Missing data were handled by listwise deletion, with no imputation performed.
- Provide complete adjusted p-value tables in the Supplementary material; pre-specify the primary endpoints.
Adjusted p-values are already presented in the text, figures, and in Supplementary Tables 3 and 5 (LME model outputs). The primary and secondary endpoints were pre-specified and are described in the Methods section (line 162 onwards). No changes were needed, but we have carefully checked the section to ensure this is clearly presented.
- Explain late-onset local reactions; give SAE/AESI narratives; and explain assessor blinding and causality assessment criteria.
Late-onset local reactions are described on Lines 250 to 256 (TC version). It concerned injection site pain and swelling, i.e. AEs considered solicited but occurring after the reporting period for solicited events (7 days). The text has been updated to clarify.
Considering all reported SAEs and AESIs were assessed as unrelated to the study vaccine, we believe including full narratives would unnecessarily lengthen supplementary materials without adding meaningful value to the conclusions. If wanted, the narratives can be provided to the reviewer.
Assessor blinding has now been added on Lines 129-136 (TC version).
Causality assessment criteria have been added on Lines 148-154 (TC version).
- Future OA cohorts should incorporate measures of cellular immunity; if at all possible, correlate HI/MN with functional assays.
We agree with the reviewer that cellular immune responses will provide important additional insights., This is discussed in the revised manuscript (Line 420 to 424 TC version). The previous study in YA identified the ability of LVA to induce influenza-specific CD4+ T cell responses and ongoing future work will investigate this in OA, along with the mechanisms of action of LVA.
- To identify asymptomatic infections, consider scheduled PCR or serology; emphasize assay validation in the body of the text.
In our phase 1a study (ref 16), serology (anti-NP antibodies) was used to determine intercurrent infections but because of NP contamination in inactivated vaccines, we were not able to accurately distinguish infection from vaccine-induced immune responses.
In the present trial participants with respiratory symptoms performed antigen self-tests which provided a rapid and pragmatic alternative to PCR. No intercurrent infections were detected. We recognize that antigen tests have lower sensitivity than PCR, and that scheduled PCR testing would likely have had limited added value in this context, as asymptomatic influenza infections are often transient and easily missed at fixed sampling points
Importantly, we did not observe subjects with unexpectedly high HI or MN titres at any timepoint, suggesting that intercurrent infections were absent or had negligible impact on the overall immune response. This limitation has now been acknowledged in the discussion section (Line 411-414, TC version).
- Increase recruitment to include frail and racially/ethnically diverse OA populations.
We fully agree on the importance of including frail and racially/ethnically diverse older adults in future studies, since these groups have higher risk for severe influenza infection. However, this study was a Phase 1b trial and was the first conducted in older adults with this adjuvant. As such, our primary objective was to assess safety in a well-characterized, healthy OA cohort to minimize confounding factors and ensure interpretability of early-phase data. Future studies, including planned Phase 2 trials, will expand recruitment to include more diverse and representative OA populations, including individuals who are frail or from racially and ethnically diverse backgrounds.
Round 2
Reviewer 2 Report
Comments and Suggestions for Authors
The authors have revised their manuscript in accordance with the reviewers' comments. However, some of the meanings in the drawings are barely discernible due to the very small font. It would be helpful if the authors enlarged the font on some of the labels in the vaccine immunogenicity drawings.
Author Response
The authors have revised their manuscript in accordance with the reviewers' comments. However, some of the meanings in the drawings are barely discernible due to the very small font. It would be helpful if the authors enlarged the font on some of the labels in the vaccine immunogenicity drawings.
Thank you again for helping to improve the manuscript. Indeed, the size reduction by introducing the figures inside the word document resulted in labels being still too small. We have increased the size of the labels but additionally rotated the figures. These are now placed in vertical order which results in a better fit on the pages in the word document. The p-values and values in the seroconversion/seroprotection plot are still smaller than the other labels but are set to their maximum size that does not result in overlap of the numbers. The quality should be enough to read the labels still. Overall, this leads to better readability of the figures.